# Large Animal Studies to Reduce the Foreign Body Reaction in Brain–Computer Interfaces: A Systematic Review

**DOI:** 10.3390/bios11080275

**Published:** 2021-08-16

**Authors:** Shan Yasin Mian, Jonathan Roy Honey, Alejandro Carnicer-Lombarte, Damiano Giuseppe Barone

**Affiliations:** 1Department of Surgery and Cancer, Faculty of Medicine, Imperial College London, London SW7 2BX, UK; 2School of Clinical Medicine, University of Cambridge, Cambridge CB3 0DF, UK; jrh210@cam.ac.uk; 3Electrical Engineering Division, Department of Engineering, University of Cambridge, Cambridge CB3 0DF, UK; ac723@cam.ac.uk; 4Centre for Brain Repair, Department of Clinical Neurosciences, University of Cambridge, Cambridge CB3 0DF, UK; dgb36@cam.ac.uk

**Keywords:** in vivo, animal, brain–computer, interface, astrogliosis

## Abstract

Brain–computer interfaces (BCI) are reliant on the interface between electrodes and neurons to function. The foreign body reaction (FBR) that occurs in response to electrodes in the brain alters this interface and may pollute detected signals, ultimately impeding BCI function. The size of the FBR is influenced by several key factors explored in this review; namely, (a) the size of the animal tested, (b) anatomical location of the BCI, (c) the electrode morphology and coating, (d) the mechanics of electrode insertion, and (e) pharmacological modification (e.g., drug eluting electrodes). Trialing methods to reduce FBR in vivo, particularly in large models, is important to enable further translation in humans, and we systematically reviewed the literature to this effect. The OVID, MEDLINE, EMBASE, SCOPUS and Scholar databases were searched. Compiled results were analysed qualitatively. Out of 8388 yielded articles, 13 were included for analysis, with most excluded studies experimenting on murine models. Cats, rabbits, and a variety of breeds of minipig/marmoset were trialed. On average, over 30% reduction in inflammatory cells of FBR on post mortem histology was noted across intervention groups. Similar strategies to those used in rodent models, including tip modification and flexible and sinusoidal electrode configurations, all produced good effects in histology; however, a notable absence of trials examining the effect on BCI end-function was noted. Future studies should assess whether the reduction in FBR correlates to an improvement in the functional effect of the intended BCI.

## 1. Introduction

The brain–computer interface (BCI), often described as a neural interface, is of growing interest to neuroscientists and clinicians, as well as material and bioelectrical engineers. Safely interfacing neurons with a sensor or probe offers exciting new avenues in the treatment of neurological disease [1,2]. In its simpler form, it is already employed through deep brain stimulation (DBS), where electrodes deliver current to a specific region of the brain, with improvements in symptoms in pathologies such as Parkinson’s disease [3]. More complex interfaces, like those in retinal and cochlear implants, can ameliorate impaired vision, and restore some function to perceptive faculties previously thought irreparable [4,5]. The scope of application is wide, demonstrated by the recent STIMO trial, where patients with partial spine transection were able to regain some motor function in their lower limbs using simple electrode stimulators applied to target spinal nerves [6]. The ability to accurately record and actuate the nervous system may create new treatment modalities, and give life to a new generation of sophisticated robotic prostheses.

Sensors, effectors, resistors, transistors and a variety of electrical components may constitute part of the BCI. In this review, we seek to evaluate the interactions of parts of this circuit with living biology. In the aforementioned context of DBS, for example, the peri-electrode parenchyma is known to undergo fibrosis, with the deposition of scar tissue on and around the electrode [7]. This leads to the development of an area of tissue with increased astrocyte presence and thickness, and subsequently the occlusion of action potential propagation due to a complex interplay of factors, including the high impedance of this tissue [8,9,10].

This phenomenon, generally referred to as the foreign body reaction (FBR) and mediated in the brain by astrogliosis, is accounted for by numerous cellular processes. Principally, an inflammatory complex is thought to form with a variety of astroglia, microglia (the resident macrophage and immune cell within the brain) and a milieu of inflammatory cytokines assembling [11]. These result in the deposition of the scar tissue over time, high in glial fibrillary acidic protein (GFAP) [12], an important structural protein in normal nervous tissue, but pathological in high concentrations

Often, the electrodes may be implanted in eloquent nervous tissue, where any deposition of scar tissue may begin to manifest debilitating neurological effects at low thresholds [13]. In addition, the fidelity of recorded electrical activity is reliant upon proximity of the probes to the tissue. BCIs aim to record nuanced, relatively weak currents, with a great number and variety of individual action potentials; to record these with sufficient accuracy requires small, delicately formed electrodes, which are easily deformed or enveloped by scar tissue [8].

Despite recent progress, obstacles remain in the translation of BCIs, which thus far have succeeded mainly in coarse electrical stimulation, such as in DBS electrodes. Fine sensor and probe technology, key to the recording of action potentials, and the evocation of perceptions that underpin a successful robotic prosthesis for example, remains less developed [14,15].

Addressing the FBR may thus be considered a key priority in advancing progress in BCIs, as ever smaller electrodes will require its attenuation to function. Significant research has been conducted on this topic, and in this systematic review, we aim to evaluate the progress made using both pharmacological and material science, in tackling this sophisticated biological phenomenon.

The FBR occurs after a foreign body, such as an electrode, is introduced into the brain tissue. Initially, the wound is just a disturbance in the local tissue architecture, but the many months process of inflammation, remodelling and change in the affected tissue is what constitutes the entire FBR. It must as a process be considered separately to the physiological process of astrogliosis, which although sharing some features, can occur without foreign bodies and hold inherently useful properties [16,17]. Astrogliosis occurs after inflammatory, autoimmune, infective and ischaemic injuries to nervous tissue, as well as trauma [18]. During inflammation in multiple sclerosis, or blood–brain barrier (BBB) breakdown in ischaemic stroke, astrogliosis can protect against further damage [19,20,21]. Evidence shows that the scar dampens the inflammatory milieu, and physically blocks further extrusion and infiltration of additional inflammatory microglia and macrophages, stabilising the inflammatory process.

With respect to the BBB, the astroglia recruited during astrogliosis are of similar morphology to those in the structure of the normal BBB. The filopodia (feet-like projections) of these cells form a rudimentary BBB, and go on to become a permanent replacement as the tissue repairs [22]. As such, when the penetrating electrodes, or those resting on the cortical surface (such as in electrocorticography, or ECoG systems) cause crush injury to the neurons, this part of astrogliosis renders the FBR as not an entirely harmful process. A balance between preserving the useful properties of astrogliosis (such as BBB repair and dampening of inflammation), while mitigating the limiting effects of the FBR overall on BCIs (such as polluting the recordings of electrical probes or diminishing the strength of stimulant electrodes on the nervous tissue) is required.

To this effect, many studies have been conducted to explore the cellular processes that underpin the FBR, as well as trial methods to reduce it. In vivo and in vitro studies demonstrate how neural tissue is reliant on regular tissue architecture and that interruption of this architecture by electrodes creates the FBR [23]. To counter this, probes constructed with flexible materials, or impregnated with cytotropic agents have demonstrated reduction of the FBR. Furthermore, porous probe materials have shown some promising results in vitro. Insertion of probes at a reduced speed also limits the traumatic impact of the probes (the so called dimpling effect), reducing the FBR [24]—although this could form standard practice for future electrode insertions, there is no in vivo evidence it would, on its own, sufficiently dampen the FBR to reduce its harmful effects.

In lieu of this, different probe surfactant types have also been used to minimise the FBR. Parylene C and Carbon/Tin alloys have demonstrated some success in vitro [25], although correct application often requires electroplating, and is resource intensive and technically difficult. The above strategies are targeted at reducing the effect of the probe on the FBR.

Pharmacologically, steroid compounds are also widely used in a variety of inflammatory processes in the brain, such as dexamethasone in oedema and inflammation caused by brain tumours. Given their efficacy, their use has also been trialled extensively in the reduction of the FBR.

In vitro, steroids were infused in growth medium containing neuronal cells and cultured against non-active probes, demonstrating dampening of the inflammatory milieu [26]. Subsequently, dexamethasone administered systemically to small animal models demonstrated significant reduction in FBR measured qualitatively on pathology [27]. Recently, drug eluting electrodes, which diffuse dexamethasone into their local surroundings, showed a reduction in the FBR without the side effects associated with systemic steroid medication that have been trialled in vivo. These showed promising results in murine models, and represent a promising development in countering FBR [28].

However, aside from steroids, there have been remarkably few compounds that have succeeded in FBR reduction. One may conjecture that recently developed biologics and disease modifying anti-rheumatics, could in the future play a part in reducing FBRs, although this remains untested [29].

Combined material modification and administration of time limited systemic therapies in the literature demonstrate that FBR attenuation can be achieved to allow BCIs to function effectively. What is noticeable is that many of these strategies thus far have been trialled in vitro, or at best in murine models, with a significant sparsity of literature that evaluates them in large animal groups.

This is significant, given that evidence already exists supporting the theory that FBR may manifest differently dependent on the size and anatomy of the implanted tissue [30], with recent electrodes getting larger and being implanted over a broader area [31]. Several FBR-attenuating techniques may work well in a rodent model, such as porous, cuff-shaped or drug-eluting electrodes; however, when trialled in a large animal, the dimensions and forces of impact between the interface of the electrode and the tissue are likely to be different, and thus they may have vastly different effects on FBR size [32]. As electrodes invariably increase in size to accommodate BCIs in larger animals and eventually humans, the methods with which to control FBR in those subjects should also be trialled in larger animals [13,33].

With this in mind, we aimed to systematically search the literature, and establish the key themes in the evidence that concern the success of reducing the FBR in the large animal model. By evaluating the state of this evidence, future goals of research can be sought with greater ease, and the potential targets for translating any success in reducing FBR to human models in the future may be considered.

## 2. Methods

We employed the PRISMA guidelines to conduct our review [34]. Literature was collated from the MEDLINE, EMBASE, SCOPUS and Scholar databases. MEDLINE and EMBASE were searched using the OVID system, with search terms and date of search indexed and recorded, and appendant. Where possible, MeSH headings were exploded, and the NOT function avoided, as we aimed to deploy a wide-ranging strategy, capturing as many articles as possible. In SCOPUS, the native SCOPUS search engine was used, with the search term also recorded, as was the case for Scholar.

All searched articles were compiled into one database using EndNote, where deduplication was conducted, first using the automated tool, and then manually by two independent authors (SM and JH). Two authors independently screened abstracts, and their results were cross-matched to ensure no discrepancies arose (SM and JH). Any discrepancy in inclusion for an article was then automatically taken forward for full text screening. Full-text screening followed the above protocol, and discrepancies in this phase were adjudicated upon by the two non-screening authors through consensus (AC and DB).

The exclusion and inclusion criteria are as follows: Studies must report the in vivo use of an approach to reducing the FBR in implants inserted into any structures that comprise the nervous system. Murine and guinea pig models were not included. Rabbits and Minipigs were permitted; however, these were the smallest models reviewed in this paper. A control group without intervention was not compulsory, and we included articles which compared against existing controls in the literature, or other BCI intervention groups without any control. However, articles reporting only one arm of data, without any comparator whatsoever were excluded. No limit was placed on date of publication, country of origin or journal. All study samples must consist of at least two subjects and provide original evidence-review articles were not considered. All studies would only have their data counted once, if reported twice (either in two separate publications, or once in abstract); the more contemporary and complete report of the data was included.

Once articles were screened, included articles had their reference lists and articles citing them retrospectively searched through SCOPUS, and screened as additions. Data collection and tabulation was conducted through Excel, and tables produced through RevMan. In addition to study demographics, we aimed to extract the locus of BCI, the electrode types, as well as the key salient conclusions drawn from the studies.

We used an adapted version of the ROBINs-1 criteria for the evaluation of within study bias. Finally, where data was missing or remained to be reported, we contacted authors to seek full disclosure.

## 3. Results

Our systematic review yielded a total of 8388 articles, with a range of journal pieces, abstracts, and other data submissions. Once deduplicated and screened, 38 articles remained. Through manual additions process, a further total of 2429 articles were added. Once these were deduplicated and screened, a combined total of 52 articles had their full texts retrieved, which were then screened. This ultimately yielded 13 articles included for analysis, with full disclosure of search results reported in the PRISMA flowchart in Figure 1. Of those articles excluded, many were literature reviews, trials using rodent models, or viability studies submitted to demonstrate new models of electrode as being safe to function or implant, often presented as technical notes rather than novel studies.

Given the breadth of variables in the included papers, for clarity our results section shall be divided into and analyse: (a) the different systems of measuring the FBR; (b) the animal tested; (c) the anatomical location of the neural interface, subdivided into the electrode’s morphology (penetrating vs. planar) and coating; (d) the mechanics of electrode insertion; and (e) any pharmacological modification, such as drug-eluting electrodes.

(a)Evaluating the FBR

With regard to outcomes, all 13 studies focused on evidence of FBR attenuation using immunohistochemistry, with a few also reporting electrophysiological outcomes. No study reported behavioural metrics, whereby the faculty that the BCI was meant to improve or restore could have been quantified.

With respect to immunohistochemistry, the techniques used were homogeneous. Perfusion was most often achieved with intracardial saline and paraformaldehyde, after anaesthetization of the animals. Subsequently, glial fibrillary acidic protein (GFAP) was stained for using primary goat anti-GFAP antibodies; in some cases these were supplemented with biotinylated rabbit anti-goat antibodies. The stain created was green in colour and examined using conventional laboratory microscopes with 20 to 40 degrees of magnification.

No formal system was devised to delineate whether a sample had undergone FBR, shown by the presence of astrogliosis and associated GFAP, with trials judging this subjectively. This was supplemented among trials with the use of scanning electron microscopy (SEM). Although SEM was most often used for the purposes of identifying electrode microstructure, its use also allowed studies to measure the length of staining around the electrode in micrometres (microns), thus along with density provided a quantitative comparison.

GFAP staining was recorded up to 800 microns away from the edges of inserted electrodes. Where a microelectrode array (MEA) was used, which consists of numerous small electrodes in close proximity to each other, staining was confined to 50 to 100 microns.

With respect to electrophysiological recordings, some articles used electrical corticography (ECoG), while others formed a compound measure of altered impedance, and recorded signal intensity from the electrodes. Unfortunately, these were highly heterogeneous, and relative ratios were most commonly described to quantify the efficacy of the FBR reduction technique on electrophysiological readings.

(b)The animals tested

With regard to species trialled, and for ease of reference, the following Latin names were applied:

*Oroctolagus cuniculus* = rabbit

*Felis catus* = domestic cat

*Sus Scrofa domesticus* = minipig

*Callithrix jacchus* = common marmoset monkey

The most commonly used species were the cat and rabbit, with the number of subjects ranging from 2 to 10 subjects per trial. The majority of trials lasted between 4 and 8 months, with the exception of rabbits, where trials on average lasted 16 months. Unfortunately, gender and other biometric data, including size or any co-morbidities within the animals were not commonly reported. Given that most trials used animals early on in their life, or those specifically bred for laboratory study, these variables were assumed to be absent.

(c)The anatomical location of the neural interface, subdivided into the electrodes’ morphology and coating

The majority of electrodes were trialled in the central nervous system (CNS), placed in the cortex, while four studies implanted on the peripheral nervous system (PNS), such as the sciatic, dorsal genital and recurrent laryngeal nerves.

With respect to electrodes in the CNS, the interventions could be broadly divided into two groups: those affecting the design of the electrode themselves, and those modifying their coating. Electrode specific interventions in the CNS included the design of a flexible, sinusoidal probe, and the use of a chiselled tip (similar to the UTAH slant tip electrode). These interventions produced a significant reduction in density of staining from 30 microns onwards to the extent of the total range (around 500 microns on average). Notably, there was no effect on the total range of staining, and density in the first 30 microns remained the same throughout the trials.

BCI coatings in the CNS included boron diamond on a carbon electrode, Teflon, polyamide, and silicon-iridium. Broadly, these had little effect, with Teflon, silicon and boron/diamond having no significant impact on the intensity or range of GFAP staining. Polyamide conversely had an adverse effect with increased density of staining throughout the 40- to 80-micron range surrounding the electrode.

Those studies referring to the PNS were similar in that they either used an alternative structure of electrode, or an alternate coating. PNS electrodes in the included studies were shaped as cuffs with implanted electrode segments on the inner surface. Two studies used alternative cuff thicknesses, and a custom cuff demonstrated increased staining when the cuff was narrower, up to a margin of 30% increased density. Furthermore, these studies also demonstrated myelin thinning, a phenomenon uniquely reported in the PNS trials. One trial used a porous tin nitride coated electrode cuff, which produced no significant impact on the FBR. Lastly, another trial compared whether the nature of distal repair of an implanted nerve (anastomosis or autograft), would affect the FBR in the proximal stimulation electrode; it too reported no significant impact

(d)The mechanics of electrode insertion

Additionally, in the CNS there were also trials which changed depth of stimulation, with middle and proximal segments of the electrodes being activated in turn. In addition, there was a trial which used a regime of low current stimulation out with the function of the electrode to see if this would affect the FBR. Changing depth had no impact; however, he regime of low current stimulation reduced the range of staining, producing a reduction of 50 microns.

(e)Pharmacological intervention

Our search revealed a paucity of research into drug eluting electrodes or the use of systemic immunomodulation in large mammals with BCIs. The next step on from murine studies, such as the use of drug elution by FitzGerald in 2016, would be to evaluate its efficacy in larger mammals [28].

With respect to bias, the modified ROBINS tool was applied to all studies, with modifications and reasoning recorded. In general, we judged there to be a high risk of bias with respect to selective reporting. This was in large part due to a large number of slices and imagery being taken in every trial, but only a few being chosen for stain density examination, microscopy and ultimately reporting. This was often done without a standardised process, and on the discretion of investigators, rendering the main scope of potential bias in these trials. Figure 2 demonstrates the sum of our assessment, showing the large skew of risk being in the domain of reporting bias.

## 4. Discussion

Our literature review, although systematic, yielded relatively few articles fitting our main objective of attenuating the FBR on neural interfaces in vivo. The creation of a well-optimised BCI is an endeavour that remains in its relative infancy, and significant further study will be required before they may be used for a restorative or enhancing intent. Although public interest grows in BCI and early studies demonstrating the success of stimulatory probes in spinal cord injury are promising, they remain exceedingly rare [35], a finding supported by the low acceptance rate of our own study, which suggests there is not enough evidence to draw any meaningful conclusions thus far. Table 1 in the results section is a summary of the studies we yielded, along with their most salient conclusions, which we will explore further here. Figure 3 later in this section represents a schematic representation of the literature, and the parts of a BCI probe that are modified to aid with ease of summary.

### 4.1. Evaluation of the FBR and Animal Size

To achieve the high fidelity BCI that is capable of more than just coarse stimulation will require an evidenced method by which to reduce the FBR, which will inevitably require trial in large animals. Previous studies demonstrate that immune reactions can vary significantly between murine and large animal models [36], and consequently methods to control them will vary in efficacy. Our initial perspective, that the FBR being strongly mediated by mechanical mismatch and movement, as well as the size of involved tissue is strengthened by the results of our review and furthers the need to see trials using models such as primates or cattle. Several viability studies on primates did feature in our non-included articles, and their reported electrophysiological recordings, SEM and immunohistochemical staining of perfused animals provide interesting insights into the FBR in larger animals. However, the lack of controls rendered them unfit for inclusion; if the trialled methods are ever to be translated into humans, then primate studies will require cohort controls [37,38].

This is further reinforced by the fact that many of these primate studies, even with short study periods, demonstrate some degradation of signal or destruction of electrode to cortex interface on SEM [8,39]. Similarly, humans’ electrodes used for DBS or laser intra-thermal therapy often require replacement or their locus changed to maintain functionality in the longer term [40].

### 4.2. Methods to Reduce the FBR in the CNS—Physiological

Turning to the content of included articles, those focusing on the CNS offered the most substantial findings. Starting with the electrode design, the first finding of note was that stimulating only one segment of the electrode did not have any effect. No other studies were found trialling this method, suggesting that all six layers of the cerebral cortex behave similarly with regard to the FBR. Given the majority of evidence indicates that the glia (microglia in particular) are most responsible for the FBR, and are distributed evenly throughout the cortex [41,42], these findings are in keeping with the current theory on FBR and depth of stimulation.

On the other hand, Lenarz et al.’s 2007 study using regimented stimulation did reduce the range of observed FBR. There is evidence that excessive current and stimulation will destroy local tissue and produce the FBR, similar to the burning effect in electrosurgical cauterization [43]. This principle is employed in a disciplined manner in laser intra-thermal therapy, to target epileptogenic tracts of nerves or tumour [44,45]. There is an argument, however, that stimulation at lower intensities, mimicking normal cortical activity as in Lenarzs study, may in fact have the inverse effect and reduce the FBR. In a way, stimulating local tissue into attenuating the FBR and recovering more quickly. Themes in the literature would support this, as otherwise inflammatory and traumatic lesions that also undergo less FBR, tend to heal better and with greater plasticity if the lesions and their surrounding tissues are stimulated [46,47]. This technique does pose obstacles for electrodes designed purely for recording, however, as unlike electrodes with a primary function of stimulation (such as DBS electrodes), passing current through electrodes reserved purely for recording may produce adverse effects. They may cause a plethora of deficits depending on the target region. As such, for the moment, the evidence from our review does not support the method of modifying local electrophysiology as a means of tackling the FBR.

### 4.3. Methods to Reduce the FBR in the CNS—Mechanical

The second category of interventions consisted of modifications in tip design, with a chisel tip electrode demonstrating reduced gliosis after the first 30 microns. Similar effects have been widely reported with slanted tips, and overall slanting in the array with sequential shortening of electrodes (such as the UTAH slant tip array) in rodent studies [15]. It is well documented that the penetrating “crush” of the electrode as it slides into position contributes significantly to the FBR that will arise [38,48,49]. Modifying electrode tips, such as the mosquito inspired microprobe insertors [50] have previously been trialled. However, the chisel tip seems to represents a simpler and less resource intensive strategy.

In addition to crush injury, the physical entry force of the electrodes in MEAs can manifest as “dimpling”. This is where multiple electrode tips can tent and irritate the dura to cause fibrosis and calcification. Although not mentioned in the articles we found, this represents another benefit of a non-flat electrode tip, as those with a slanted profile are likely to cause less dimpling during penetration, thereby reducing the cumulative FBR.

The theory behind why a reduced crush effect from a narrower tip would reduce FBR also explains why the sinusoidal electrodes demonstrated some success in reducing FBR. CNS tissue is averse to trauma, as demonstrated by diffusion tract imaging and diffuse axonal injury, where inertia and energy injuries through concussion imparted on the tissue can cause diffuse inflammation, and subsequently gliosis similar to the FBR [51]. In BCI electrodes there is a recognised motion effect, where the host’s natural movements cause the MEA, or singular electrodes, and the tissue they are implanted in to move against each other. Although the structures involved are small, the friction and shear forces imparted by the relatively rigid electrodes on the surrounding tissue can worsen the FBR, particularly over time

The sinusoidal probe trialled by Sohal et al. is a development of previous flexible electrodes trialled in vitro, and its success in mitigating the FBR is unsurprising [51]. The altered shape of the electrode sits better in the host tissue, and aids in transmitting the forces of movement more evenly, causing less shear stress on the surrounding tissues.

Ultimately, a probe would have to make its way into the cortex to realise the BCI. Some techniques trialled previously, including allowing the probe to “grow” into the cortex, or create a coating that is in essence bio-hybrid [52], may not always permit the succinctly implantable BCIs that scientists aim to create. As such, some minimal crush injury and “dimpling” will remain an accepted risk [53], and the FBR incurred as a result of this may be unavoidable. Despite this, it is evident from the findings of our review, that the use of malleable, smaller and possibly more numerous electrodes in the CNS holds some promise in the reduction of the FBR going forward.

### 4.4. Methods to Reduce the FBR in the CNS—Coatings

This brings us on to the final theme identified among the papers studying the CNS that we returned, which was the coating of the electrodes. Coating the electrodes is different from plating: plating being a process by which separate metals are made part of the electrode structure; whereas a coating is more loosely adhered and can erode or turn away from the surface with greater ease. In our literature search, Teflon, Silicon/Iridium oxide, polyamide and boron/diamond on a carbon base were identified as potential covering agents. The biology that governs how the FBR in neural tissue behaves on different surfaces is beyond the scope of this work, and is subject to a variety of different processes.

The first compound trialled as a coating was a polyamide. In this case, the polyamide coating in fact led to an increase in measured FBR against the control. While alternative polyamide coatings are available, including aromatic and aliphatic variants, our evidence did not suggest any utility of these compounds in countering the FBR.

Teflon similarly resulted in a largely detrimental effect, with a range of gliosis of 1000 microns in the primate model compared to rodent averages of 800 microns used as a comparator-whether this difference would be innate to the size and local biology of the primate is unclear. Despite Teflon being widely employed for its properties of inertness, and mitigating friction and interaction between synthetic compounds, in the in vivo setting it did not seem to have this effect.

This left the metallic plating and coatings within our search. In the case of silicon and iridium oxide used for plating in Han et al.’s 2012 study, this had minimal impact on the outcomes. There was only a slight improvement in electrophysiological recordings in cats and the range and density of gliosis in the rabbit studies comparable with other studies examining rabbit FBR. The properties of silicon have previously been studied with in vitro BCIs, and their touted efficacy as conductors may explain why the improvement in recordings in the cat trials were noted [54].

The final coating in question for the CNS papers was the boron/diamond coated carbon electrode. Unfortunately, the substantial rabbit study did not demonstrate any significant difference in FBR on pathology, although a very slight improvement in electrophysiological readings was noted over the course of the study. Overall, the coating thus far in the CNS, according to our evidence do not particularly affect the FBR, with material selection and configuration appearing to be of greater significance.

### 4.5. Methods to Reduce the FBR in the PNS—Mechanical

Turning our attention to the articles studying the PNS, one ought to preface the evaluation by understanding the differences in FBR, and the factors affecting BCI design in the PNS. Anatomically, given the mammalian PNS is usually at its greatest diameter at the sciatic nerve, it is still much narrower than the majority of structures in the CNS, hence creating implantable electrodes is generally difficult. This is compounded by the fact that the previously mentioned shear forces that nerves can undergo, which can affect the junction between the electrode and the nerve, can in peripheral nerves go as far as to dislodge the implanted electrode entirely. This has happened in cat trials previously, as the location of the sciatic nerve in the hind limb is particularly prone to motions that can dislodge the electrodes [55].

The cuff system can preclude the electrode from detaching, while the penetrating components of the array on the interior keep it in place from sliding along the length of the nerve. Problems remain however, as penetrating electrodes tend to be placed along the entirety of the inside of the cuff, hence selective recording/stimulation remains difficult. This is particularly pertinent in large nerves, which contain components of several downstream branches hence making focal recording and stimulation an important faculty to consider [37,56].

To this end, the Leventhal et al. 2006 study developed the flat interface nerve electrode (FINE) cuff. This cuff was of ovoid shaped, and aimed to alter the shape of the nerve, and let it be compressed such that selective stimulation would be possible, as well as a reduced number of penetrating electrodes overall as only the necessary segments would be implanted, thereby reducing the cumulative FBR. Although a reduction of FBR was not demonstrated with use of the cuff, cuff tightness did demonstrate an increase in FBR. This is in line with previously discussed theory on the crush effect of electrodes, and in this case the cuff. Another additional phenomenon noted here was “myelin thinning”, which was not commented upon as one of the sequelae in the CNS studies.

Rigidity modification also features in our review, with Zhen et al. in 2019 trialling a custom flexible electrode, also in cuff form, on the sciatic nerve of rabbits. Although in the PNS, this interestingly did not lead to a reduction in gliosis density and the long-term signal recorded from the flexible cuffs were of greater quality. This again is in line with the CNS flexible electrode findings, which suggests that the benefit of the flexibility is in the long term, as natural micromotions have less of a cumulative effect in the build-up of the FBR. Myelin thinning however was also noted in this study, and so gives further weight to the theory that it may not actually be a result of the crushing effect of the cuff alone.

As with the CNS, the articles studying the PNS demonstrate that modifying the mechanics of the BCI probe insertion, and using malleable materials, can be used to reduce the FBR.

### 4.6. Methods to Reduce the FBR in the PNS—Coating and Summary

Coatings were also applied in the PNS studies, with porous tin nitride plating used in Meijs et al.’s 2016 study on minipigs. The porous nature of the plating aims to allow the nerve to grow into the medium, allowing the tissue to become part of the electrode surface with greater ease than if it was smooth and flat. No noticeable reduction in gliosis was noted. Aside, from this, there was a surprising sparsity of literature on the use of coatings in the PNS, and no evidence to suggest any work so far was found.

The final PNS study we yielded used a surprising experimental paradigm to investigate the FBR. Smith et al. in 2012 in their cat study placed the stimulant electrode in the recurrent laryngeal nerve of cats whose distal nerve segment had been repaired either with direct anastomosis or a graft nerve from a different part of the cat’s body (autograft). In both cases, no difference in the extent of FBR was found. This demonstrated that the method by which the damaged nerve is repaired prior to connection with a BCI, has no significant impact on the extent of FBR attenuation.

As to the problem of detaching electrodes and micromotions inducing FBR, a phenomenon that was remarked upon in several PNS studies, some rodent study trials previously used electrodes that are inserted longitudinally rather than perpendicularly to the course of the nerve. These electrodes, in essence insert along single or multiple individual bundles or neurolemmas, and do not necessarily cross the nerve [57,58]. This permits an increased surface area of recording and also creates a greater area of traction, thereby reducing the risk of the electrode being removed during natural motion. Further exploration in large animal models is needed [59]. This is also the case for the trial of electrode coatings, which did not generally produce reduction in the FBR. Myelin thinning, one of the unexpected sequelae reported, is detrimental to nerve functioning [55,60] and future efforts to tackle it may be key to also reducing the FBR in electrodes implanted in the PNS.

Across both PNS and CNS, the subgroups of electrode coatings and configuration have featured most commonly, with systemic FBR reduction therapies featuring less so. Although interesting in its own right, it also fits with the evidence from murine studies, and it is surprising that studies did not start ahead with experimenting in these domains from the outset. Mesh type devices, and those that are softer and offer greater conformity, have already been evidenced to demonstrate reduction in FBR [61,62,63]; hence, using electrodes that are described by our yielded studies as still being “rigid”, even on a control basis does not seem worthwhile.

Furthermore, there is longstanding evidence demonstrating that shear forces may increase the FBR, particularly in modalities where the electrodes are tethered to the skull [64]. Similarly, we suggested here an electrode that may grow or become inherent to the local tissue architecture, as an alternative to the current insertable types that are available. This too has already previously been used in a viability study, using silk meshes and capillary forces to enable resorption [65]. Unfortunately, the use of these techniques to aid with compliance in the tissue were either not readily described, or well considered in the articles we yielded. This was one of the key limitations of our review, and when the time to freely trial implantable electrodes in humans on a large scale arises, then these technical factors must be considered. To start again from scratch with rigid, non-mesh type devices from the outset would render an unnecessary delay. Finally, these factors must also be described in greater detail, and ought to be considered in future reviews as the evidence base grows greater, as only through a multi-factorial approach such as this can the overall problems associated with FBR be truly tackled.

## 5. Conclusions

The objective of our review was to characterise the state of evidence in large animal models of methods by which to reduce the FBR in electrodes or probes associated with BCIs. Our main finding in this respect, was that the research has not yet translated to large animals in substantial quantity, and remains confined to murine and in vitro studies thus far. Furthermore, the studies demonstrated a distinct set of subgroups in this field of research, by which future studies or reviews may be more easily categorized.

Out of the studies that do exist, the application of flexible probes reduced the FBR to a greater extent than the variety of coatings trialled. In either case however, there was insufficient evidence to conclude whether this would translate to larger animals such as primates, and eventually to humans.

Although the number of viability studies in primates for novel BCIs is promising, research in FBR optimisation evidently still lags behind. Although public interest in BCIs, typified by popular companies such as Elon Musks Neuralink [35] is at a high point, realising such high fidelity BCIs will require attenuation of the FBR, which will inevitably be preceded by large animal studies. This also holds true for alternative electrode types, such as Musks woven mesh cortical surface electrodes, or recently trialled intravascular variants, which will also need further testing in larger animals [66].

Although some may argue that a base level of FBR is acceptable, such as in DBS electrodes that are only replaced every few years, there is as of yet insufficient evidence to say if this will hold true as BCIs develop.

In lieu of this, we draw perhaps our safest conclusion, which is that future research on FBR reduction in BCI ought not only to be directed towards large animals, but must include functional outcomes. Histology, although of interest in establishing mechanisms of immune response, may have no bearing on the final utility of a BCI. If for example, a BCI with a FBR reducing measure, produces greater target response over a prolonged period compared to a control, then this is the most significant measure by which to judge that FBR reducing intervention. Electrophysiology readings are a good measure of this, and it is reassuring to see them reported regularly as outcomes in the articles yielded in our review.

The future of FBR reducing measures in BCI will be dependent upon the clinical benefit and advantage it can confer to the implants, and large animal modelling will be a key stepping stone in this venture.

## Figures and Tables

**Figure 1 biosensors-11-00275-f001:**
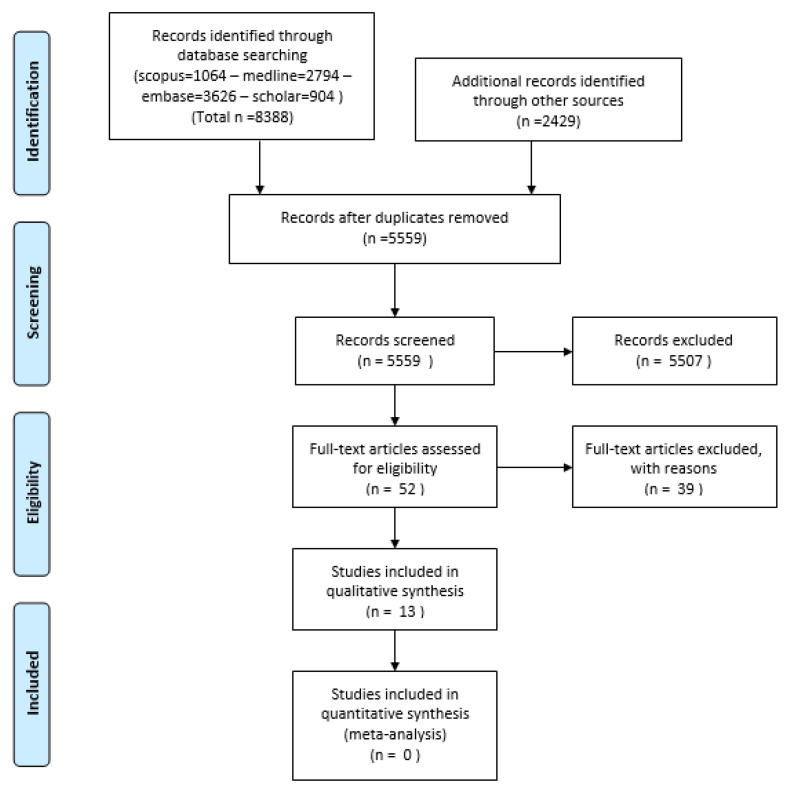
PRISMA Flowchart of analysed studies.

**Figure 2 biosensors-11-00275-f002:**
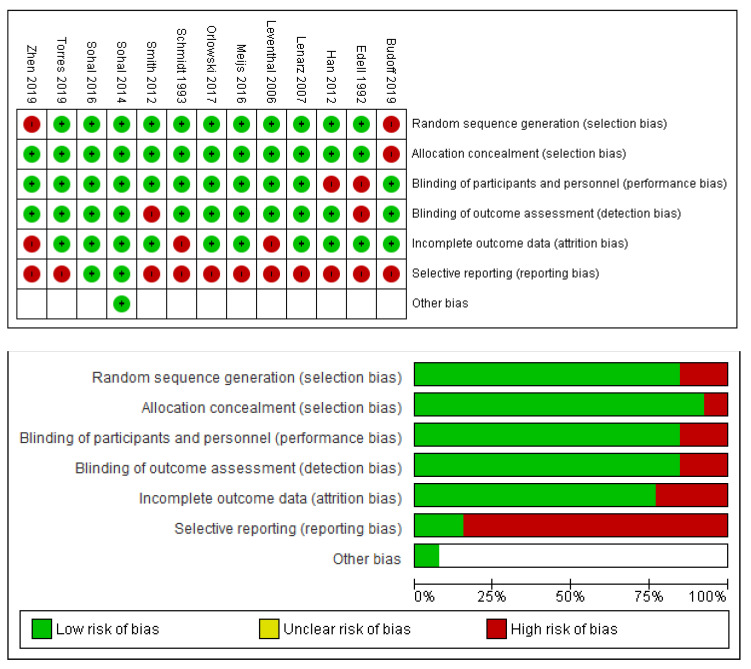
ROBINS summary charts.

**Figure 3 biosensors-11-00275-f003:**
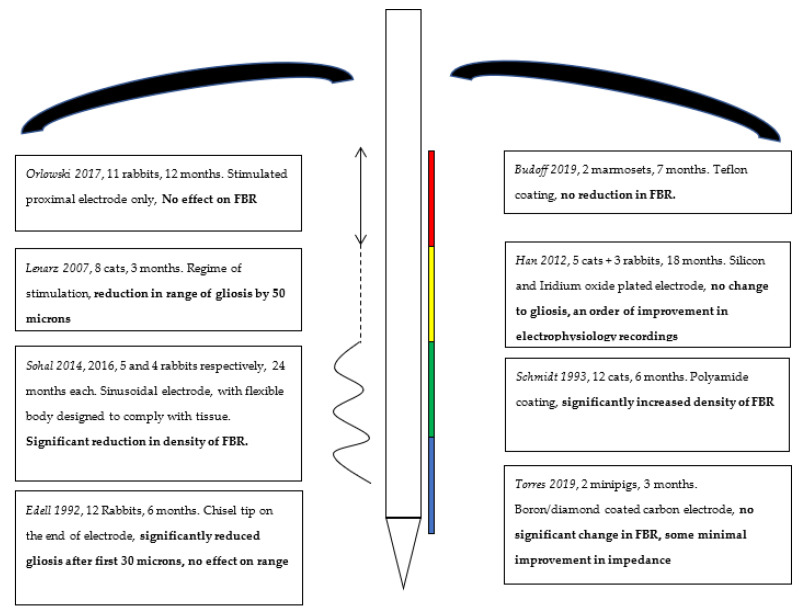
Schematic summary of interventions, coatings on right and electrode structure modifications on left, with conclusions highlighted in bold.

**Table 1 biosensors-11-00275-t001:** Summary of included articles. * are articles studying cortical implants.

Study	Species	Locus	Sample Size/Length of Study	Intervention	Control	Outcome(s)	Conclusion
Budoff 2019 *	Callithrix Jacchus	Cortex	2, 7 Months	Teflon electrode coating	Comparison made to the same electrode applied in rodent	Immunohistochemistry	Mean presence of Gliosis extends to 1000 microns, as opposed to 800 in rodents
Edell 1992 *	Oructolagus Cuniculus	Cortex	12, 6 Months	Chisel tip electrode	Control electrode inserted in same subject at different locus	Immunohistochemistry	All subjects showed significantly reduced Gliosis after first 30 Microns compared to control, extent of Gliosis was equivalent
Han 2012 *	Felis Catus/ Oructolagus Cuniculus	Cortex + VCN	5C, 3R 18 Months	Silicon + Iridium Oxide plated electrode	Control electrodes in same subject at contralateral locus	Immunohistochemistry + Electrophysiology	- Rabbits showed gliosis up to 50 microns - Cats showed an order of magnitude of improved electrophysiology recordings in intervention group (exact micron range not reported)
Lenarz 2007	Felis Catus	Inferior Colliculi	8, 3 Months	4 Hourly daily stimulation	Control group without stimulation	Immunohistochemistry	- Significant Gliosis at only 50 microns in intervention group, up to 100 in control - Some gliosis recorded up to 300 microns away in control, only 250 microns in intervention group
Leventhal 2006	Felis Catus	Sciatic Nerve	12, 3 Months	FINE Ovoid Cuff (wide, medium and narrow configurations)	Control electrode inserted at contralateral side	Immunohistochemistry	30% increase in Gliosis in narrow cuff, medium and wide comparable
Meijs 2016	Sus Scrofa Domesticus	Dorsal Genital Nerve	2, 2 Months	Porous Tin Nitride plated electrode	Control electrode inserted at contralateral side	Immunohistochemistry	No significant change in Gliosis
Orlowski 2017 *	Sus Scrofa Domesticus	Cortex	11, 12 Months	Altering depth of stimulation segment	Control inserted contralaterally, different depths activated	Immunohistochemistry	No significant changes in Gliosis with varying height, mean range was 500 microns
Schmidt 1993 *	Felis Catus	Cortex	12, 6 Months	Polyamide coating of electrode	Control uncoated electrodes inserted contralaterally	Immunohistochemistry	Polyamide coating significantly increased Gliosis density, with no effect on range of around 40–80 microns
Smith 2012	Felis Catus	RLGN	12, 3 Months	Electrode effect measured in anastomosed and autografted nerves	Controlled against similar trials in the literature	Immunohistochemistry	No significant change in Gliosis, either against controls or in method of repair
Sohal 2016 *	Oructolagus Cuniculus	Cortex	4, 24 Months	Sinusoidal, flexible electrode	Controlled against traditional, rigid electrode	Immunohistochemistry	Significant reduction in Gliosis up to 500 microns, no change in range of Gliosis observed between groups
Sohal 2014 *	Oructolagus Cuniculus	Cortex	5, 24 Months	Sinusoidal, flexible electrode	Controlled against traditional, rigid electrode	Immunohistochemistry + Electrophysiology	- Significant reduction in Gliosis up to 500 microns - Significantly increased signal intensity in the sinusoidal probes
Torres 2019 *	Sus Scrofa Domesticus	Cortex	2, 3 Months	Boron/Diamond coated Carbon electrode	Controlled against traditional Platinum Iridium electrode	Immunohistochemistry + Electrophysiology	- No significant change in Gliosis between control and intervention electrodes - Non-significant improvement in impedance in the intervention electrodes
Zhen 2019	Oructolagus Cuniculus	Sciatic Nerve	14, 9 Months	Novel custom electrode	Controlled against traditional straight electrode, and sham cuff	Immunohistochemistry + Electrophysiology	- Reduced electrophysiology recordings at 2 months, but improved by 6 months and after - No significant difference in Gliosis, but there was notable myelin thinning throughout

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
