# Peer review of "Large Animal Studies to Reduce the Foreign Body Reaction in Brain–Computer Interfaces: A Systematic Review"

_biosensors, 2021, doi:10.3390/bios11080275_

Round 1
Reviewer 1 Report
Overall I found the work well done, clearly written, and useful to the community. Below I add some minor points to address.
- P3: Line 25-26: I don’t think this is true. Brain surgeries in humans can take hours, and if slowly inserting electrodes was beneficial, this could certainly be implemented in humans.
- P4: Line 14: Could be good to cite the Neuropixels paper as an example of electrodes getting larger.
- P7: Line 15: “Broadly, these had little effect, with Teflon, Silicon and Boron/diamond significant impact on the intensity or range of GFAP staining” This is confusing. Did it have little effect? Or significant impact?
- P8: Han 2012 conclusion: “Cats showed an order of improved electrophysiology recordings in intervention group” Order of magnitude?
- P8 table: Align to left, rather than justify the text
- It would be good to reference the table and figures in the main text.
- Figure 3: Missing text
- Page 16: Lines 32-36: revise phrases
- Page 17: Line 7: “number of viability of studies”. Please revise
- Perhaps the discussion could include novel promising technologies? Such as intravascular electrodes (Fisher Nat Biom Eng 2018), neural dust, or Musk’s sewn electrodes.
Reviewer 2 Report
Firstly, this review addresses a very critical area in BCI research, and is a very appropriate topic to address! The authors have used very rigorous metrics to search for literature, and highlighted the relative dearth of publications on this topic.
I have a couple of comments for the authors to consider. Both are related to the organization and representation of data:
1) The collection and organization of source literature, and the prospect of bias in those papers is well characterized. This is very helpful.
2) There is a lot of heterogeneity in the types of implantable probes in the cited studies. One particularly meaningful distinction is CNS vs PNS electrodes. And within the CNS group, planar ECoG type electrodes vs intracortical penetrating electrodes (Utah array type). The current organization of the paper would benefit from explicit declaration of these features and categorization of results into CNS/PNS, epi/intracortical categories, particularly, for example in Table 1 which surveys all included studies.
3) The authors note that the quantification of FBR is either histological or visual SEM etc. However, it would be very helpful in a review article to have these outcomes tabulated in a quantitative way.
Reviewer 3 Report
Summary: the authors present a review of the foreign body reaction in the brain computer interfaces. Although it is difficult to follow their methodologies. The authors screened over 8,000 publications and only included 13. The authors arrive at disparate conclusions and state that further research including functional outcomes is in order. The reviewer has both general and specific comments for the authors.
GENERAL COMMENTS: Manuscript is too long and needs to be shorten considerable
In the intro the authors should summarize the current state of the literature and discuss future developments. The manuscript often rambles of topic and does not appear to follow a logical sequence. This manuscript is difficult to follow as it is not well constructed.
SPECIFIC COMMENTS:
ABSTRACT: Very superficial abstract stating a number of compilations of different topics with limited connections
Needs a transition to explain why the need for larger animal studies; why the large jump from 8388 to 13 articles? This needs a short explanation.
INTRODUCTION: The introduction should summarize the parameters of the FBR with respect to the brain computer interface. Comments regarding peripheral nerves are not germane to the discussion. is too long. Limit the introduction to the foreign body reaction
METHODS: This needs to be rewritten such that any other investigator would be able to reproduce the search strategy.
The inclusion and exclusion criteria should be written and not provided in a bulleted list
Why do the authors mention meta-analysis as this was neither the intend of this manuscript nor do the authors include any publications?
Authors state restriction to large animals but include rats and cats into the analysis. The authors should rather state the animals they focus on.
RESULTS: This needs to be shortened considerable. The results should address the issues raised in the method section.
Page 6 line 12: Felis catus
DISCUSSION: This needs to be shortened considerable. The authors should limit their comments to what was reported in the result section and speculations should be kept to a minimum.
Page 5 of 20 figure 3: much of the text in the boxes has been truncated
Reviewer 4 Report
The authors of this review picked a very interesting topic in the field of neural interfaces. However, the way the review process was approached is debatable.
The stability of issue-electrode interface is critical to the success of a reliable neuroscientific study and of the clinical translation of such neural technologies. The authors acknowledged that and rightfully spoke about FBR as one of the main causes of electrode failure.
Rather than using an algorithm based on keywords for the general collection of the literature to review, they could have structured the search into 'subgroup' where the main contributing factors to FBR were discussed (e.g., implantation speed of intracortical implants, structural biocompatibility of implants, corrosion of the electrodes). I believe that by choosing the more general approach in their search, the could have missed some important manuscripts on this topic.
In particular, on the structural biocompatibility (biomechanics interaction) of the implanted devices, they should elaborate on the approaches used by the reviewed literature (substrate material used, design of the implant, implantation setup etc..). This is a topic of great relevance but, in order for the translation among species to successfully happen, the critical factors to increase tissue compliance should be identified and tailored according to the application. This aspect is practically missing in this review, and it should be added to increase the value of the manuscript.
Some relevant literature which the authors should consider reviewing and citing/including here:
- 10.1088/1741-2560/11/4/046011 and other publications from this group
- https://doi.org/10.1002/jbm.a.31138 and other publications from this group
- doi.org/10.1016/j.biomaterials.2020.120178 and other publications from this group
- doi.org/10.1002/adma.201906512 and other publications from this group
- doi.org/10.1038/nmat2745
The research listed above provides very useful insights on technical aspects directly correlated to the longevity of the implants and the way they function in vivo. Please integrate and use those researches to discuss the matter of structural biocmopatibility.
Other comments:
(Abstract): What do the authors mean with 'subject size'? can electrodes and coatings be considered techniques? The authors state that studies on large animals are required. This is true only but to increase the chances of success much more has to be understood (or incorporated) at a technology level. I think this concept should be clarified. What are functional trials?
(Introduction): Second paragraph, neurological diseases instead of diseases. Page 2, authors state that glial scar formation leads to neuronal death (elaborate on this, how?) and inhibits propagation of action potentials. I think this is incorrect, glial scar encapsulates the electrodes and prevents high-quality recordings when it becomes thicker but does not necessarily means that the action potentials are inhibited. Please explain and provide supporting literature.
The authors talk about 'deposition of scar tissue', but I assume they mean 'formation, growth'? They also state that 'FBR occurs when a foreign body is introduced into the brain'. By only introducing an object in the brain/body, a would is created (wound healing cascade initiates). FBR is related to the long-term effects to the presence of a device in the brain/other parts of the body. Please check on biological responses and time-related phases following an implantation.
What are "nervous implants"? Probably not the right terminology..
Page 6: Last large paragraph gives details about the staining and scarring but what type of implants and materials were considered? Unfortunately, this is where I see the lack of structure in this review: the boundary conditions of the studies reported here are not always clear and it is easy to get confused about the context.
I agree with the authors on that very often (too often) the extraordinary technological advancement reported in literature does not correspond to real advancement on the application forefront but there are still groups focused on the fundamental issues of tissue-electrode interface dynamics and biomechanics and I believe this review should focus on putting those aspect together before even going into the large-animals studies (and in that part, they should use a more defined structure).
Round 2
Reviewer 3 Report
The article's title implies that the article focuses on the foreign body reaction in brain computer interfaces. While this subject is addressed, other spurious and tangential items are included in the manuscript. The manuscript is lengthy and could be shortened considerably with minimal impact on content. The specific comments are as follows:
- Introduction- This should summarize the literature. The authors stated that there is a paucity of information on this subject and yet their introduction exceeds three pages.
- Method section-This includes a flow chart that should be included in the result section. Notably, the species evaluated should be included in this section but were included in the result section.
- Results - the flow chart, which was placed in the method section indicates that only 13 of 8,388 publications were included in the analysis. This is an acceptance rate of 0.15%. Therefore, either their method or their search strategy was specious.
- Discussion - this needs to be shortened considerably. The authors need to state what the manuscript contributes to the literature.
Reviewer 4 Report
Thanks to the authors for providing answers to my questions. I am pleased with the majority of the implemented changes.
I understand that it is not suitable to re-do the literature survey at this point, thus, if the editor/s agree, this review can be published as is.
Author Response
Dear Reviewer,
Many thanks for your kind words. We have taken aboard the lessons learnt, and will apply them thoroughly as we continue our groups work in this field. Many thanks again for your feedback, and we look forward to sharing the published manuscript in the next issue of Biosensors.Yours faithfully,
S Y Mian
Corresponding author